# CONTROL-AWARE REPRESENTATIONS FOR MODEL-BASED REINFORCEMENT LEARNING

**Brandon Cui**
Facebook AI Research
bcui@fb.com

**Yinlam Chow**
Google Research
yinlamchow@google.com

**Mohammad Ghavamzadeh**
Google Research
ghavamza@google.com

## ABSTRACT

A major challenge in modern reinforcement learning (RL) is efficient control of dynamical systems from high-dimensional sensory observations. Learning controllable embedding (LCE) is a promising approach that addresses this challenge by embedding the observations into a lower-dimensional latent space, estimating the latent dynamics, and utilizing it to perform control in the latent space. Two important questions in this area are how to learn a representation that is amenable to the control problem at hand, and how to achieve an end-to-end framework for representation learning and control. In this paper, we take a few steps towards addressing these questions. We first formulate a LCE model to learn representations that are suitable to be used by a policy iteration style algorithm in the latent space. We call this model *control-aware representation learning* (CARL). We derive a loss function and three implementations for CARL. In the *offline* implementation, we replace the locally-linear control algorithm (e.g., iLQR) used by the existing LCE methods with a RL algorithm, namely model-based soft actor-critic, and show that it results in significant improvement. In *online* CARL, we interleave representation learning and control, and demonstrate further gain in performance. Finally, we propose *value-guided* CARL, a variation in which we optimize a weighted version of the CARL loss function, where the weights depend on the TD-error of the current policy. We evaluate the proposed algorithms by extensive experiments on benchmark tasks and compare them with several LCE baselines.

## 1 INTRODUCTION

Control of non-linear dynamical systems is a key problem in control theory. Many methods have been developed with different levels of success in different classes of such problems. The majority of these methods assume that a model of the system is known and its underlying state is low-dimensional and observable. These requirements limit the usage of these techniques in controlling dynamical systems from high-dimensional raw sensory data (e.g., image), where the system dynamics is unknown, a scenario often seen in modern reinforcement learning (RL).

Recent years have witnessed a rapid development of a large arsenal of model-free RL algorithms, such as DQN (Mnih et al., 2013), TRPO (Schulman et al., 2015), PPO (Schulman et al., 2017), and SAC (Haarnoja et al., 2018), with impressive success in solving high-dimensional control problems. However, most of this success has been limited to simulated environments (e.g., computer games), mainly due to the fact that these algorithms often require a large number of samples from the environment. This restricts their applicability in real-world physical systems, for which data collection is often a difficult process. On the other hand, model-based RL algorithms, such as PILCO (Deisenroth & Rasmussen, 2011), MBPO (Janner et al., 2019), and Visual Foresight (Ebert et al., 2018), despite their success, still face difficulties in learning a model (dynamics) in a high-dimensional (pixel) space.

To address the problems faced by model-free and model-based RL algorithms in solving high-dimensional control problems, a class of algorithms have been developed, whose main idea is to first learn a low-dimensional latent (embedding) space and a latent model (dynamics), and then use this model to control the system in the latent space. This class has been referred to as *learning controllable embedding* (LCE) and includes algorithms, such as E2C (Watter et al., 2015), RCE (Banijamali et al., 2018), SOLAR (Zhang et al., 2019), PCC (Levine et al., 2020), Dreamer (Hafner et al., 2020a;b), PC3 (Shu et al., 2020), and SLAC (Lee et al., 2020). The following two properties are extremely important in designing LCE models and algorithms. **First**, to learn a representation that is the most suitable for the control problem at hand. This suggests incorporating the control algorithm in the

process of learning representation. This view of learning control-aware representations is aligned with the value-aware and policy-aware model learning, VAML (Farahmand, 2018) and PAML (Abachi et al., 2020), frameworks that have been recently proposed in model-based RL. **Second**, to interleave the representation learning and control, and to update them both, using a unifying objective function. This allows to have an end-to-end framework for representation learning and control.

LCE methods, such as SOLAR, Dreamer, and SLAC, have taken steps towards the second objective by performing representation learning and control in an online fashion. This is in contrast to offline methods like E2C, RCE, PCC, and PC3 that learn a representation once and then use it in the entire control process. On the other hand, methods like PCC and PC3 address the first objective by adding a term to their representation learning loss function that accounts for the curvature of the latent dynamics. This term regularizes the representation towards smoother latent dynamics, which are suitable for the locally-linear controllers, e.g., iLQR (Li & Todorov, 2004), used by these methods.

In this paper, we take a few steps towards the above two objectives. We first formulate a LCE model to learn representations that are suitable to be used by a policy iteration (PI) style algorithm in the latent space. We call this model *control-aware representation learning* (CARL) and derive a loss function for it that exhibits a close connection to the prediction, consistency, and curvature (PCC) principle for representation learning (Levine et al., 2020). We derive three implementations of CARL: *offline*, *online*, and *value-guided*. Similar to offline LCE methods, such as E2C, RCE, PCC, and PC3, in *offline* CARL, we first learn a representation and then use it in the entire control process. However, in offline CARL, we replace the locally-linear control algorithm (e.g., iLQR) used by these LCE methods with a PI-style (actor-critic) RL algorithm. Our choice of RL algorithm is the model-based implementation of soft actor-critic (SAC) (Haarnoja et al., 2018). Our experiments show significant performance improvement by replacing iLQR with SAC. *Online* CARL is an iterative algorithm in which at each iteration, we first learn a latent representation by minimizing the CARL loss, and then perform several policy updates using SAC in this latent space. Our experiments with online CARL show further performance gain over its offline version. Finally, in *value-guided* CARL (V-CARL), we optimize a weighted version of the CARL loss function, in which the weights depend on the TD-error of the current policy. This would help to further incorporate the control algorithm in the representation learning process. We evaluate the proposed algorithms by extensive experiments on benchmark tasks and compare them with several LCE baselines: PCC, SOLAR, and Dreamer.

## 2 PROBLEM FORMULATION

We are interested in learning control policies for non-linear dynamical systems, where the states $s \in \mathcal{S} \subseteq \mathbb{R}^{n_s}$ are not fully observed and we only have access to their high-dimensional observations $x \in \mathcal{X} \subseteq \mathbb{R}^{n_x}$, $n_x \gg n_s$. This scenario captures many practical applications in which we interact with a system only through high-dimensional sensory signals, such as image and audio. We assume that the observations $x$ have been selected such that we can model the system in the observation space using a Markov decision process (MDP)[1] $\mathcal{M}_{\mathcal{X}} = \langle \mathcal{X}, \mathcal{A}, r, P, \gamma \rangle$, where $\mathcal{X}$ and $\mathcal{A}$ are observation and action spaces; $r : \mathcal{X} \times \mathcal{A} \to \mathbb{R}$ is the reward function with maximum value $R_{\max}$, defined by the designer of the system to achieve the control objective;[2] $P : \mathcal{X} \times \mathcal{A} \to \mathbb{P}(\mathcal{X})$ is the unknown transition kernel; and $\gamma \in (0, 1)$ is the discount factor. Our goal is to find a mapping from observations to control signals, $\mu : \mathcal{X} \to \mathbb{P}(\mathcal{A})$, with maximum expected return, i.e., $J(\mu) = \mathbb{E}[\sum_{t=0}^{\infty} \gamma^t r(x_t, a_t) \mid P, \mu]$.

Since the observations $x$ are high-dimensional and the observation dynamics $P$ is unknown, solving the control problem in the observation space may not be efficient. As discussed in Section 1, the class of *learning controllable embedding* (LCE) algorithms addresses this by learning a low-dimensional latent (embedding) space $\mathcal{Z} \subseteq \mathbb{R}^{n_z}$, $n_z \ll n_x$, together with a latent dynamics, and controlling the system there. The main idea behind LCE is to learn an *encoder* $E : \mathcal{X} \to \mathbb{P}(\mathcal{Z})$, a *latent space dynamics* $F : \mathcal{Z} \times \mathcal{A} \to \mathbb{P}(\mathcal{Z})$, and a *decoder* $D : \mathcal{Z} \to \mathbb{P}(\mathcal{X})$,[3] such that a good or optimal controller (policy) in $\mathcal{Z}$ performs well in the observation space $\mathcal{X}$. This means that if we model the control problem in $\mathcal{Z}$ as a MDP $\mathcal{M}_{\mathcal{Z}} = \langle \mathcal{Z}, \mathcal{A}, \bar{r}, F, \gamma \rangle$ and solve it using a model-based RL algorithm to obtain a policy $\pi : \mathcal{Z} \to \mathbb{P}(\mathcal{A})$, the image of $\pi$ back in the observation space, i.e.,

---

[1]A method to ensure observations are Markovian is to buffer them for several time steps (Mnih et al., 2013).

[2]For example, in a goal tracking problem in which the agent (robot) aims at finding the shortest path to reach the observation goal $x_g$ (the observation corresponding to the goal state $s_g$), we may define the reward for each observation $x$ as the negative of its distance to $x_g$, i.e., $-\|x - x_g\|^2$.

[3]Some recent LCE models, such as PC3 (Shu et al., 2020), are advocating latent models without a decoder. Although we are aware of the merits of such approach, we use a decoder in the models proposed in this paper.

---

**Algorithm 1** Latent Space Learning with Policy Iteration (LSLPI)

---

1: **Inputs**: $E^{(0)}$, $F^{(0)}$, $D^{(0)}$;
2: **Initialization:** $\mu^{(0)} = $ random policy; $\quad$ $\mathcal{D} \leftarrow$ samples generated from $\mu^{(0)}$;
3: **for** $i = 0, 1, \ldots$ **do**
4: $\quad$ Compute $\pi^{(i)}$ as the projection of $\mu^{(i)}$ in the latent space w.r.t. $D_{\text{KL}}\big(\pi \circ E \ || \ \mu\big)$; # $\mu^{(i)} \approx \pi^{(i)} \circ E^{(i)}$
5: $\quad$ Compute the value function of $\pi^{(i)}$ and set $V^{(i)} = V_{\pi^{(i)}}$; $\qquad$ *# policy evaluation (critic)*
6: $\quad$ Compute the greedy policy w.r.t. $V^{(i)}$ and set $\pi_+^{(i)} = \mathcal{G}[V^{(i)}]$; $\qquad$ *# policy improvement (actor)*
7: $\quad$ Set $\mu^{(i+1)} = \pi_+^{(i)} \circ E^{(i)}$; $\qquad$ *# project the improved policy $\pi_+^{(i)}$ back into the observation space*
8: $\quad$ Learn $(E^{(i+1)}, F^{(i+1)}, D^{(i+1)}, \bar{r}^{(i+1)})$ from $\mathcal{D}$, $\pi^{(i)}$, and $\pi_+^{(i)}$; $\qquad$ *# representation learning*
9: $\quad$ Generate samples $\mathcal{D}^{(i+1)} = \{(x_t, a_t, r_t, x_{t+1})\}_{t=1}^n$ from $\mu^{(i+1)}$; $\quad$ $\mathcal{D} \leftarrow \mathcal{D} \cup \mathcal{D}^{(i+1)}$;
10: **end for**

---

$(\pi \circ E)(a|x) = \int_z dE(z|x)\pi(a|z)$, should have high expected return. Thus, the loss function to learn $\mathcal{Z}$ and $(E, F, D)$ from observations $\{(x_t, a_t, r_t, x_{t+1})\}$ should be designed to comply with this goal.

This is why in this paper, we propose a LCE framework that tries to incorporate the control algorithm used in the latent space in the representation learning process. We call this model, *control-aware representation learning* (CARL). In CARL, we set the class of control (RL) algorithms used in the latent space to approximate policy iteration (PI), and more specifically to soft actor-critic (SAC) (Haarnoja et al., 2018). Before describing CARL in details in the following sections, we present a number of useful definitions and notations here.

For any policy $\mu$ in $\mathcal{X}$, we define its value function $U_\mu$ and Bellman operator $T_\mu$ as

$$U_\mu(x) = \mathbb{E}[\sum_{t=0}^\infty \gamma^t r_\mu(x_t) \mid P_\mu, x_0 = x], \qquad T_\mu[U](x) = \mathbb{E}_{x' \sim P_\mu(\cdot|x)}[r_\mu(x) + \gamma U(x')], \quad (1)$$

for all $x \in \mathcal{X}$ and $U : \mathcal{X} \to \mathbb{R}$, where $r_\mu(x) = \int_a d\mu(a|x)r(x, a)$ and $P_\mu(x'|x) = \int_a d\mu(a|x)P(x'|x, a)$ are the reward function and dynamics induced by $\mu$. Similarly, for any policy $\pi$ in $\mathcal{Z}$, we define its induced reward function and dynamics as $\bar{r}_\pi(z) = \int_a d\pi(a|z)\bar{r}(z, a)$ and $F_\pi(z'|z) = \int_a d\pi(a|z)F(z'|z, a)$. We also define its value function $V_\pi$ and Bellman operator $T_\pi$ as

$$V_\pi(z) = \mathbb{E}[\sum_{t=0}^\infty \gamma^t \bar{r}_\pi(z_t) \mid F_\pi, z_0 = z], \qquad T_\pi[V](z) = \mathbb{E}_{z' \sim F_\pi(\cdot|z)}[\bar{r}_\pi(z) + \gamma V(z')]. \quad (2)$$

For any policy $\pi$ and value function $V$ in the latent space $\mathcal{Z}$, we denote by $\pi \circ E$ and $V \circ E$, their image in the observation space $\mathcal{X}$, given encoder $E$, and define them as

$$(\pi \circ E)(a|x) = \int_z dE(z|x)\pi(a|z), \qquad\qquad (V \circ E)(x) = \int_z dE(z|x)V(z). \quad (3)$$

## 3 CARL MODEL: A CONTROL PERSPECTIVE

In this section, we formulate our LCE model, which we refer to as *control-aware representation learning* (CARL). As described in Section 2, CARL is a model for learning a low-dimensional latent space $\mathcal{Z}$ and the latent dynamics, from data generated in the observation space $\mathcal{X}$, such that this representation is suitable to be used by a policy iteration (PI) style algorithm in $\mathcal{Z}$. In order to derive the loss function used by CARL to learn $\mathcal{Z}$ and its dynamics, i.e., $(E, F, D, \bar{r})$, we first describe how the representation learning can be interleaved with PI in $\mathcal{Z}$. Algorithm 1 contains the pseudo-code of the resulting algorithm, which we refer to as *latent space learning policy iteration* (LSLPI).

Each iteration $i$ of LSLPI starts with a policy $\mu^{(i)}$ in the observation space $\mathcal{X}$, which is the mapping of the improved policy in $\mathcal{Z}$ in iteration $i - 1$, i.e., $\pi_+^{(i-1)}$, back in $\mathcal{X}$ through the encoder $E^{(i-1)}$ (Lines 6 and 7). We then compute $\pi^{(i)}$, the current policy in $\mathcal{Z}$, as the image of $\mu^{(i)}$ in $\mathcal{Z}$ through the encoder $E^{(i)}$ (Line 4). Note that $E^{(i)}$ is the encoder learned at the end of iteration $i - 1$ (Line 8). We then use the latent space dynamics $F^{(i)}$ learned at the end of iteration $i - 1$ (Line 8), and first compute the value function of $\pi^{(i)}$ in the policy evaluation or *critic* step, i.e., $V^{(i)} = V_{\pi^{(i)}}$ (Line 5), and then use $V^{(i)}$ to compute the improved policy $\pi_+^{(i)}$, as the greedy policy w.r.t. $V^{(i)}$,

i.e., $\pi^{(i+1)} = \mathcal{G}[V^{(i)}]$, in the policy improvement or *actor* step (Line 6). Using the samples in the buffer $\mathcal{D}$, together with the current policies in $\mathcal{Z}$, i.e., $\pi^{(i)}$ and $\pi_+^{(i)}$, we learn the new representation $(E^{(i+1)}, F^{(i+1)}, D^{(i+1)}, \bar{r}^{(i+1)})$ (Line 8). Finally, we generate samples $\mathcal{D}^{(i+1)}$ by following $\mu^{(i+1)}$, the image of the improved policy $\pi_+^{(i)}$ back in $\mathcal{X}$ using the old encoder $E^{(i)}$ (Line 7), and add it to the buffer $\mathcal{D}$ (Line 9), and the algorithm iterates. It is important to note that both critic and actor operate in the low-dimensional latent space $\mathcal{Z}$.

LSLPI is a PI algorithm in $\mathcal{Z}$. However, what is desired is that it also acts as a PI algorithm in $\mathcal{X}$, i.e., it results in (monotonic) policy improvement in $\mathcal{X}$, i.e., $U_{\mu^{(i+1)}} \geq U_{\mu^{(i)}}$. Therefore, we define the representation learning loss function for CARL, such that it ensures LSLPI also results in policy improvement in $\mathcal{X}$. The following theorem, whose proof is reported in Appendix A, shows the relationship between the value functions of two consecutive polices generated by LSLPI in $\mathcal{X}$.

**Theorem 1.** *Let $\mu$, $\mu_+$, $\pi$, $\pi_+$, and $(E, F, D, \bar{r})$ be the policies $\mu^{(i)}$, $\mu^{(i+1)}$, $\pi^{(i)}$, $\pi_+^{(i)}$, and the learned latent representation $(E^{(i+1)}, F^{(i+1)}, D^{(i+1)}, \bar{r}^{(i+1)})$ at iteration $i$ of the LSLPI algorithm (Algorithm 1). Then, the following holds for the value functions of $\mu$ and $\mu_+$:*

$$U_{\mu_+}(x) \geq U_\mu(x) - \Big( \frac{1}{1-\gamma} \sum_{\widetilde{\pi} \in \{\pi, \pi_+\}} \mathbb{E}_{d_{\widetilde{\pi} \circ E}^\gamma} [\Delta(E, F, D, \bar{r}, \widetilde{\pi}, \cdot)|x_0 = x]$$
$$+ \frac{\sqrt{2}\gamma R_{\max}}{1-\gamma} \cdot \mathbb{E}_{d_{\pi \circ E}^\gamma} [\underbrace{\sqrt{D_{KL}\big((\pi \circ E)(\cdot'|\cdot) \,||\, \mu(\cdot'|\cdot)\big)}}_{\mathrm{L_{reg}}(\mathrm{E}, \mu, \pi, \cdot)} |x_0 = x] \Big), \tag{4}$$

*for all $x \in \mathcal{X}$, where $d_{\pi \circ E}^\gamma(x'|x_0) = (1-\gamma) \cdot \sum_{\ell=0}^\infty \gamma^\ell \mathbb{P}(x_\ell = x'|x_0; \pi \circ E)$ is the $\gamma$-stationary distribution induced by policy $\pi \circ E$, and the error term $\Delta$ for a policy $\pi$ is given by*

$$\Delta(E, F, D, \bar{r}, \pi, x) = \frac{R_{\max}}{1-\gamma} \overbrace{\sqrt{\frac{-1}{2} \int_z dE(z|x) \log D(x|z)}}^{\mathrm{(I)} = \mathrm{L_{ed}}(\mathrm{E}, \mathrm{D}, \mathrm{x})} + 2 \overbrace{\Big| r_{\pi \circ E}(x) - \int_z dE(z|x) \bar{r}_\pi(z) \Big|}^{\mathrm{(II)} = \mathrm{L_r}(\mathrm{E}, \bar{r}, \pi, \mathrm{x})} \tag{5}$$
$$+ \frac{\gamma R_{\max}}{\sqrt{2}(1-\gamma)} \Big( \underbrace{\sqrt{D_{KL}\big(P_{\pi \circ E}(\cdot|x) \,||\, (D \circ F_\pi \circ E)(\cdot|x)\big)}}_{\mathrm{(III)} = \mathrm{L_p}(\mathrm{E}, \mathrm{F}, \mathrm{D}, \pi, \mathrm{x})} + \underbrace{\sqrt{D_{KL}\big((E \circ P_{\pi \circ E})(\cdot|x) \,||\, (F_\pi \circ E)(\cdot|x)\big)}}_{\mathrm{(IV)}} \Big).$$

It is easy to see that LSLPI guarantees (policy) improvement in $\mathcal{X}$, if the terms in the parentheses on the RHS of (4) are zero. We now describe these terms. The last term on the RHS of (4) is the KL between $\pi^{(i)} \circ E$ and $\mu^{(i)} = \pi^{(i)} \circ E^{(i)}$. This term can be seen as a regularizer to keep the new encoder $E$ close to the old one $E^{(i)}$. The four terms in (5) are: **(I)** The encoding-decoding error to ensure $x \approx (D \circ E)(x)$; **(II)** The error that measures the mismatch between the reward of taking action according to policy $\pi \circ E$ at $x \in \mathcal{X}$, and the reward of taking action according to policy $\pi$ at the image of $x$ in $\mathcal{Z}$ under $E$; **(III)** The error in predicting the next observation through paths in $\mathcal{X}$ and $\mathcal{Z}$. This is the error between $x'$ and $\hat{x}'$ shown in Fig. 1(a); and **(IV)** The error in predicting the next latent state through paths in $\mathcal{X}$ and $\mathcal{Z}$. This is the error between $z'$ and $\tilde{z}'$ shown in Fig. 1(b).

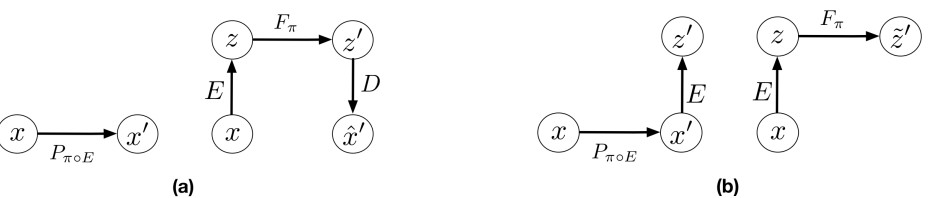

Figure 1: **(a)** Paths from the current observation $x$ to the next one, (left) in $\mathcal{X}$ and (right) through $\mathcal{Z}$. **(b)** Paths from the current observation $x$ to the next latent state, (left) through $\mathcal{X}$ followed by encoding and (right) starting with encoding and then through $\mathcal{Z}$.

**Representation Learning in CARL**   Theorem 1 provides us with a recipe (loss function) to learn the latent space $\mathcal{Z}$ and $(E, F, D, \bar{r})$. In CARL, we propose to learn a representation for which the terms in the parentheses on the RHS of (4) are small. As mentioned earlier, the second term,

$L_{\text{reg}}(E, \mu, \pi, x)$, can be considered as a regularizer to keep the new encoder $E$ close to the old one $E_-$, when the policy $\mu$ is given by $\pi \circ E_-$. Term (I) minimizes the reconstruction error between encoder and decoder, which is standard for training auto-encoders (Kingma & Welling, 2013). Term (II) that measures the mismatch between rewards can be kept small, or even zero, if the designer of the system selects the rewards in a compatible way[4]. Although CARL allows us to learn a reward function in the latent space, similar to several other LCE works (Watter et al., 2015; Banijamali et al., 2018; Levine et al., 2020; Shu et al., 2020), in this paper, we assume that a compatible latent reward function is given. Terms (III) and (IV) are the equivalent of the *prediction* and *consistency* terms in PCC (Levine et al., 2020) for a particular latent space policy $\pi$. Since PCC has been designed for an offline setting (i.e., one-shot representation learning and control), its prediction and consistency terms are independent of a particular policy and are defined for state-action pairs. While CARL is designed for an online setting (i.e., interleaving representation learning and control), and thus, its loss function at each iteration depends on the current latent space policies $\pi$ and $\pi_+$. As we will see in Section 4, in our offline implementation of CARL, these two terms are similar to prediction and consistency terms in PCC. Note that (IV) is slightly different than the consistency term in PCC. However, if we upper-bound it using Jensen inequality: $(IV) \leq L_c(E, F, \pi, x) := \int_{x' \in \mathcal{X}} dP_{\pi \circ E}(x'|x) \cdot D_{\text{KL}}(E(\cdot|x') \| (F_\pi \circ E)(\cdot|x))$, the resulted loss, $L_c(E, F, \pi, x)$, would be similar to the consistency term in PCC. Similar to PCC, we also add a curvature loss to the loss function of CARL to encourage having a smoother latent space dynamics $F_\pi$. Putting all these terms together, we obtain the following loss function for CARL:

$$\min_{E, F, D} \sum_{x \sim \mathcal{D}} \lambda_{\text{ed}} L_{\text{ed}}(E, D, x) + \lambda_{\text{p}} L_{\text{p}}(E, F, D, \pi, x) + \lambda_{\text{c}} L_{\text{c}}(E, F, \pi, x)$$
$$+ \lambda_{\text{cur}} L_{\text{cur}}(F, \pi, x) + \lambda_{\text{reg}} L_{\text{reg}}(E, \mu, \pi, x),$$
(6)

where $(\lambda_{\text{ed}}, \lambda_{\text{p}}, \lambda_{\text{c}}, \lambda_{\text{cur}}, \lambda_{\text{reg}})$ are hyper-parameters[5] of the algorithm, $(L_{\text{ed}}, L_{\text{p}})$ are the encoding-decoding and prediction losses defined in (5), $L_{\text{c}}$ is the consistency loss defined above, $L_{\text{cur}} = \mathbb{E}_{x,u}[\mathbb{E}_\epsilon [f_{\mathcal{Z}}(z + \epsilon_z, u + \epsilon_u) - f_{\mathcal{Z}}(z, u) - (\nabla_z f_{\mathcal{Z}}(z, u) \cdot \epsilon_z + \nabla_u f_{\mathcal{Z}}(z, u) \cdot \epsilon_u)\|_2^2] \mid E]$ is the curvature loss that regulates the 2$^{\text{nd}}$ derivative of $f_{\mathcal{Z}}$, the mean of latent dynamics $F$, in which $\epsilon_z, \epsilon_u$ are standard Gaussian noise, and $L_{\text{reg}}$ is the regularizer that ensures the new encoder remains close to the old one.

## 4 DIFFERENT IMPLEMENTATIONS OF CARL

The CARL loss function in (6) introduces an optimization problem that takes a policy $\pi$ in $\mathcal{Z}$ as input and learns a representation suitable for its evaluation and improvement. To optimize this loss in practice, similar to the PCC model (Levine et al., 2020), we define $\widehat{P} = D \circ F_\pi \circ E$ as a latent variable model that is factorized as $\widehat{P}(x_{t+1}, z_t, \hat{z}_{t+1}|x_t, \pi) = \widehat{P}(z_t|x_t)\widehat{P}(\hat{z}_{t+1}|z_t, \pi)\widehat{P}(x_{t+1}|\hat{z}_{t+1})$, and use a variational approximation to the interactable negative log-likelihood of the loss terms in (6). The variational bounds for these terms can be obtained similar to Eqs. 6 and 7 in Levine et al. (2020). Below we describe three instantiations of the CARL model in practice. Implementation details can be found in Algorithm 2 in Appendix D. Although CARL is compatible with most PI-style (actor-critic) RL algorithms, we choose soft actor-critic (SAC) (Haarnoja et al., 2018) as its control algorithm. Since most actor-critic algorithms are based on first-order gradient updates, as discussed in Section 3, we regularize the curvature of the latent dynamics $F$ (see Eqs. 8 and 9 in Levine et al. 2020) in CARL to improve its empirical stability and performance in policy learning.

**1. Offline CARL** We first implement CARL in an offline setting, where we generate a (relatively) large batch of observation samples $\{(x_t, a_t, r_t, x_{t+1})\}_{t=1}^N$ using an exploratory (e.g., random) policy. We then use this batch to optimize the CARL's loss function (6) via the variational approximation scheme described above, and learn a latent representation $\mathcal{Z}$ and $(E, F, D)$. Finally, we solve the decision problem in $\mathcal{Z}$ using a model-based RL algorithm, which in our case is model-based SAC[6]. The learned policy $\hat{\pi}^*$ in $\mathcal{Z}$ is then used to control the system from observations as $a_t \sim (\hat{\pi}^* \circ E)(\cdot|x_t)$. This is the setting that has been used in several recent LCE works, such as E2C (Watter et al., 2015), RCE (Banijamali et al., 2018), PCC (Levine et al., 2020), and PC3 (Shu et al., 2020). Our offline

---

[4]For example, in goal-based RL problems, a compatible reward function can be the one that measures the negative distance between a latent state and the image of the goal in the latent space.

[5]Theorem 1 provides a high-level guideline for selecting the hyper-parameters of the loss function: $\lambda_{\text{ed}} = 2R_{\max}/(1-\gamma)^2$, $\lambda_{\text{c}} = \lambda_{\text{p}} = \sqrt{2}\gamma R_{\max}/(1-\gamma)^2$, and $\lambda_{\text{reg}} = \sqrt{2}\gamma R_{\max}/(1-\gamma)$.

[6]By model-based SAC, we refer to learning a latent policy with SAC using synthetic trajectories generated by unrolling the learned latent dynamics model $F$, similar to the MBPO algorithm (Janner et al., 2019).

implementation is different than those in which **1)** we replace their locally-linear control algorithm, namely iterative LQR (iLQR) (Li & Todorov, 2004), with model-based SAC, which results in significant performance improvement, as shown in Section 5, and **2)** we optimize the CARL loss function, that despite close connection, is still different than the one used by PCC.

The CARL loss function presented in Section 3 has been designed for an online setting in which at each iteration, it takes a policy as input and learns a representation that is suitable for evaluating and improving this policy. However, in the offline setting, the learned representation should be good for any policy generated in the course of running the PI-style control algorithm. Therefore, we marginalize out the policy from the (online) CARL's loss function and use the RHS of the following corollary (proof in Appendix B) to construct the CARL loss function used in our offline experiments.

**Corollary 2.** *Let $\mu$ and $\mu_+$ be two consecutive policies in $\mathcal{X}$ generated by a PI-style control algorithm in the latent space constructed by $(E, F, D, \bar{r})$. Then, the following holds for the value functions of $\mu$ and $\mu_+$, where $\Delta$ is defined by (5) (in modulo replacing sampled action $a \sim \pi \circ E$ with action $a$):*

$$U_{\mu_+}(x) \geq U_\mu(x) - \frac{2}{1-\gamma} \cdot \max_{x, \in \mathcal{X}, a \in \mathcal{A}} \Delta(E, F, D, \bar{r}, a, x), \ \forall x \in \mathcal{X}. \tag{7}$$

**2. Online CARL** In the online implementation of CARL, at each iteration $i$, the current policy $\pi^{(i)}$ is the improved policy of the last iteration, $\pi_+^{(i-1)}$. We first generate a relatively (to offline CARL) small batch of samples using the image of the current policy in $\mathcal{X}$, i.e., $\mu^{(i)} = \pi^{(i)} \circ E^{(i-1)}$, and then learn a representation $(E^{(i)}, F^{(i)}, D^{(i)})$ suitable for evaluating and improving the image of $\mu^{(i)}$ in $\mathcal{Z}$ under the new encoder $E^{(i)}$. This means that with the new representation, the current policy that was the image of $\mu^{(i)}$ in $\mathcal{Z}$ under $E^{(i-1)}$, should be replaced by its image $\pi^{(i)}$ under the new encoder, i.e., $\pi^{(i)} \circ E^{(i)} \approx \mu^{(i)}$. In online CARL, we address this by the following *policy distillation* step in which we minimize the following loss:[7]

$$\pi^{(i)} \in \arg\min_\pi \sum_{x \sim \mathcal{D}} D_{\mathrm{KL}}\big((\pi \circ E^{(i)})(\cdot|x) \ || \ (\pi_+^{(i-1)} \circ E^{(i-1)})(\cdot|x)\big). \tag{8}$$

After the current policy $\pi^{(i)}$ is set, we perform multiple steps of (model-based) SAC in $\mathcal{Z}$ using the current model, $(F^{(i)}, \bar{r}^{(i)})$, and then send the resulting policy $\pi_+^{(i)}$ to the next iteration.

**3. Value-Guided CARL (V-CARL)** While Theorem 1 shows that minimizing the loss in (6) guarantees performance improvement, this loss does not contain any information about the performance of the current policy $\mu$, and thus, the LCE model trained with this loss may have low accuracy in regions of the latent space that are crucial for learning good RL policies. In V-CARL, we tackle this issue by modifying the loss function in a way that the resulted LCE model has more accuracy in regions with higher anticipated future returns.

To derive the V-CARL's loss function, we use the variational model-based policy optimization (VMBPO) framework by Chow et al. (2020) in which the *optimal* dynamics for model-based RL can be expressed in closed-form as $P^*(x'|x, a) = P(x'|x, a) \cdot \exp\big(\frac{\tau}{\gamma}(r(x, a) + \gamma \tilde{U}_\mu(x') - \tilde{W}_\mu(x, a))\big)$, where $\tilde{U}_\mu(x) := \frac{1}{\tau} \log \mathbb{E}\big[\exp\big(\tau \sum_{t=0}^\infty \gamma^t r_{\mu,t}\big) | P_\mu, x_0 = x\big]$ and $\tilde{W}_\mu(x, a) := r(x, a) + \frac{\gamma}{\tau} \log \mathbb{E}_{x' \sim P(\cdot|x,a)}[\exp(\tau U_\mu(x'))]$ are the *optimistic* value and action-value functions[8] of policy $\mu$, and $\tau > 0$ is a temperature parameter. Note that in the VMBPO framework, the optimal dynamics $P^*$ is value-aware, because it re-weighs $P$ with an *exponential-twisting* weight $\exp(\frac{\tau}{\gamma}w(x, a, x'))$, where $w(x, a, x') := r(x, a) + \gamma \tilde{U}_\mu(x') - \tilde{W}_\mu(x, a)$ is the temporal difference (TD) error.

In V-CARL, we use the VMBPO framework to modify the CARL's prediction loss $L_\mathrm{p}(E, F, D, \pi, x)$. Since the regularizer loss $L_\mathrm{reg}(E, \mu, \pi, x)$ in CARL forces policies $\pi \circ E$ and $\mu$ to be close to each other, we may replace the transition dynamics $P_{\pi \circ E}$ with $P_\mu$ in $L_\mathrm{p}$. This makes minimizing $L_p$ equivalent to maximizing the log-likelihood $\int_{x'} dP_\mu(x'|x) \cdot \log(D \circ F_\pi \circ E)(x'|x)$. Finally, we replace $P_\mu$ with $P_\mu^*$ in this log-likelihood and obtain $\int_a d\mu(a|x) \int_{x'} dP(x'|x, a) \cdot \exp(\frac{\tau}{\gamma} \cdot w(x, a, x')) \cdot \log(D \circ F_\pi \circ E)(x'|x)$, which is a weighted (by the exponential TD $w(x, a, x')$) log-likelihood function (w.r.t. $P$). Note

---

[7]Our experiments reported in Appendix F.1 show that adding distillation improves the performance in online CARL. Thus, all our results for online CARL and V-CARL, unless mentioned, are with policy distillation.

[8]We refer to $\tilde{U}_\mu$ as the *optimistic* value function (Ruszczyński & Shapiro, 2006), because it models the right tail of the return via the exponential utility $\rho_\tau(U(\cdot)|x, a) = \frac{1}{\tau} \log \mathbb{E}_{x' \sim P(\cdot|x, a)}[\exp(\tau \cdot U(x'))]$.

that this weight depends on the optimistic value functions $\tilde{U}_\mu$ and $\tilde{W}_\mu$. When $\tau > 0$ is small (see Appendix C for more details), these value functions can be approximated by their standard counterparts, i.e., $\tilde{U}_\mu(x) \approx U_\mu(x)$ and $\tilde{W}_\mu(x, a) \approx W_\mu(x, a) := r(x, a) + \int_{x'} dP(x'|x, a)U_\mu(x')$, which can be further approximated by their latent-space counterparts, i.e., $U_\mu(x) \approx (V_\pi \circ E)(x)$ and $W_\mu(x, a) \approx (Q_\pi \circ E)(x, a)$, according to Lemma 5 in Appendix A.1. Since the latent reward function $\bar{r}$ is defined such that $r(x, a) \approx (\bar{r} \circ E)(x, a)$, we may write the TD-error $w(x, a, x')$ in terms of the encoder $E$ and the latent value functions as $\hat{w}(x, a, x') := \int_{z, z'} dE(z|x) \cdot dE(z'|x') \cdot (\bar{r}(z, a) - Q_\pi(z, a) + \gamma V_\pi(z'))$.

## 5 EXPERIMENTAL RESULTS

In this section, we experiment with the following continuous control domains: (i) Planar System, (ii) Inverted Pendulum (Swingup), (iii) Cartpole, (iv) Three-link Manipulator (3-Pole), and compare the performance of our CARL algorithms with three LCE baselines: PCC (Levine et al., 2020), SOLAR (Zhang et al., 2019), SLAC (Lee et al., 2020), and two implementations of Dreamer (Hafner et al., 2020a) (described below).[9] These tasks have underlying start and goal states that are "not" observable, instead, the algorithms only have access to the start and goal observations. We report the detailed setup of the experiments in Appendix E, in particular, the description of the domains in Appendix E.1 and the implementation of the algorithms in Appendix E.3.

To evaluate the performance of the algorithms, similar to Levine et al. (2020), we report the %-time spent in the goal. The initial policy that is used for data generation is uniformly random (see Appendix E.2 for more details). To measure performance reproducibility for each experiment, we (i) train 25 models, and (ii) perform 10 control tasks for each model. For SOLAR, due to its high computation cost, we only train and evaluate 10 different models. Besides the average results, we also report the results from the best LCE models, averaged over the 10 control tasks.

**General Results**  Table 1 shows the means and standard errors of %-time spent in goal, averaged over all models and control tasks, and averaged over all control tasks for the best model. To compare data efficiency, we also report the number of samples required to train the latent space and controller in each algorithm. We also show the training curves (performance vs. number of samples) of the algorithms in Fig. 2. We report more experiments and ablation studies in Appendix F.

Below summarizes our main observations of the experiments. **First,** offline CARL that uses model-based SAC as its control algorithm achieves significantly better performance than PCC that uses iLQR in all tasks. This can be attributed to the advantage that SAC is more robust and effective in non-(locally)-linear environments. We report more detailed comparison between PCC and offline CARL in Appendix F.3, where we explicitly compare their control performance and latent representation maps. **Second,** in all tasks, online CARL is more data-efficient than its offline counterpart, i.e., it achieves similar or better performance with fewer samples. In particular, online CARL is notably superior in Planar, Cartpole, and Swingup, in which it achieves similar performance to offline CARL with 2, 2.5, and 4 times less samples, respectively (see Fig. 2). In Appendix F.3, we show how the latent representation of online CARL progressively improves through the iterations of the algorithm (in particular, see Fig. 11). **Third,** in the simpler tasks (Planar, Swingup, Cartpole), V-CARL performs even better than online CARL. This corroborates our hypothesis that CARL can achieve extra improvement when its LCE model is more accurate in the regions of the latent space with higher temporal difference (regions with higher anticipated future return). In 3-pole, the performance of V-CARL is worse than online CARL. This is likely due to the instability in representation learning resulted from sample variance amplification by the exponential-TD weight. **Fourth,** SOLAR requires significantly more samples to learn a reasonable latent space for control, and with limited data it fails to converge to a good policy. Even with the fine-tuned latent space from Zhang et al. (2019), its performance is incomparable to those of CARL variants and Dreamer. We report more experiments with SOLAR in Appendix F.5, in which we show that SOLAR can perform better, especially in Planar when we fix the start and goal locations. However, the improved performance is still incomparable with those of CARL and Dreamer. **Fifth,** we include an ablation study in Appendix F.2 to demonstrate how each term of the CARL's loss function impacts policy learning. It shows the importance of the prediction and consistency terms, without which the resulting algorithms struggle, and the (relatively) minor role of the curvature and encoder-decoder terms in the performance of the algorithms.

---

[9]We did not include E2C and RCE in our experiments, because Levine et al. (2020) has previously shown that PCC outperforms them.

**Dreamer** As described in Section 2, most LCE algorithms, including E2C, PCC, and CARL variants, assume the observation space $\mathcal{X}$ is selected such that the system is Markovian there. In contrast, Dreamer does not make this assumption and has been designed for more general class of control problems that can be modeled as POMDPs. Thus, it is expected that it performs inferior (requires more samples to achieve the same performance) to CARL when the system is Markov in the observation space. Moreover, CARL and other LCE methods define the reward as the negative distance to the goal in the latent space. This cannot be done in Dreamer, where the encoder is an RNN that takes an entire observation trajectory as input. To address this, we propose two methods to train the Dreamer's reward function in the latent space, which we refer to as Dreamer Pixel and Dreamer Oracle. While Dreamer Pixel uses the negative distance to the goal in the observation space $\mathcal{X}$ as the signal to train the reward function, Dreamer Oracle uses the negative distance in the (unobserved) underlying state space $\mathcal{S}$. Thus, it is more fair to compare the CARL algorithms with Dreamer Pixel than Dreamer Oracle that has the advantage of having access to the underlying state space (see Appendix F.6 for more details). As it was expected, our results show that although both Dreamer's implementations learn reasonably-performing policies for most tasks (except Planar), they require twice to 100-times more samples to achieve the same performance as the CARL algorithms. We report longer (more samples) experiments with Dreamer on all tasks in Appendix F.6 (Fig. 12).

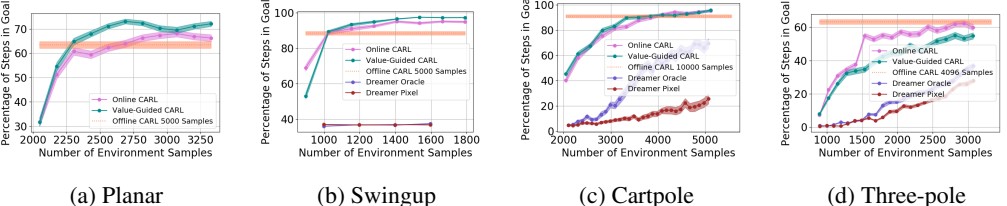

|         (a) Planar         |        (b) Swingup        |        (c) Cartpole        |        (d) Three-pole        |
| :---: | :---: | :---: | :---: |

Figure 2: Training curves of offline CARL, online CARL, V-CARL, and two implementations of Dreamer. The shaded region represents mean $\pm$ standard error.

| Environment | Algorithm | Number of Samples | Avg %-Goal | Best %-Goal |
| --- | --- | --- | --- | --- |
| Planar | PCC | 5000 | $38.85 \pm 2.45$ | $62.5 \pm 10.42$ |
| Planar | Offline CARL | 5000 | $63.43 \pm 2.78$ | $\mathbf{79.51 \pm 0.38}$ |
| Planar | Online CARL | 3072 | $68.03 \pm 1.69$ | $79.02 \pm 0.38$ |
| Planar | V-CARL | 3200 | $\mathbf{71.05 \pm 1.46}$ | $\mathbf{79.51 \pm 0.38}$ |
| Planar | SOLAR | 5000 (VAE) + 16000 (Control) | $5.82 \pm 2.50$ | $9.13 \pm 3.54$ |
| Swingup | PCC | 5000 | $86.60 \pm 1.00$ | $97.40 \pm 0.61$ |
| Swingup | Offline CARL | 5000 | $88.43 \pm 2.02$ | $\mathbf{98.50 \pm 0.0}$ |
| Swingup | Online CARL | 1408 | $95.04 \pm 0.96$ | $\mathbf{98.50 \pm 0.0}$ |
| Swingup | V-CARL | 1408 | $\mathbf{96.50 \pm 0.25}$ | $\mathbf{98.50 \pm 0.0}$ |
| Swingup | SOLAR | 5200 (VAE) + 40000 (Control) | $16.1 \pm 0.69$ | $22.45 \pm 1.96$ |
| Swingup | Dreamer Pixel | 180895 | $70.35 \pm 0.62$ | $\mathbf{98.5 \pm 0.0}$ |
| Swingup | Dreamer Oracle | 183084 | $94.65 \pm 0.20$ | $98.25 \pm 0.0$ |
| Cartpole | PCC | 10000 | $83.64 \pm 0.63$ | $\mathbf{100.0 \pm 0.0}$ |
| Cartpole | Offline CARL | 10000 | $91.11 \pm 1.50$ | $\mathbf{100.0 \pm 0.0}$ |
| Cartpole | Online CARL | 5120 | $95.34 \pm 1.17$ | $\mathbf{100.0 \pm 0.0}$ |
| Cartpole | V-CARL | 5120 | $95.79 \pm 1.06$ | $\mathbf{100.0 \pm 0.0}$ |
| Cartpole | SOLAR | 5000 (VAE) + 40000 (Control) | $10.61 \pm 2.58$ | $12.33 \pm 2.96$ |
| Cartpole | Dreamer Pixel | 96941 | $95.59 \pm 3.77$ | $\mathbf{100.0 \pm 0.0}$ |
| Cartpole | Dreamer Oracle | 14474 | $\mathbf{97.77 \pm 1.525}$ | $\mathbf{100.0 \pm 0.0}$ |
| Three-pole | PCC | 4096 | $4.41 \pm 0.75$ | $36.20 \pm 7.06$ |
| Three-pole | Offline CARL | 4096 | $63.20 \pm 1.77$ | $88.55 \pm 0.0$ |
| Three-pole | Online CARL | 2944 | $62.17 \pm 2.28$ | $\mathbf{90.05 \pm 0.0}$ |
| Three-pole | V-CARL | 2816 | $55.06 \pm 2.42$ | $89.05 \pm 0.0$ |
| Three-pole | SOLAR | 2000 (VAE) + 20000 (Control) | $0 \pm 0$ | $0 \pm 0$ |
| Three-pole | Dreamer Pixel | 6245 | $61.93 \pm 2.30$ | $\mathbf{90.00 \pm 0.0}$ |
| Three-pole | Dreamer Oracle | 6245 | $\mathbf{71.07 \pm 2.45}$ | $88.40 \pm 0.0$ |

Table 1: Mean $\pm$ standard error results (%-goal) and samples used for different LCE algorithms.

**Results with Environment-biased Sampling** In the previous experiments, all the online LCE algorithms are warm-started with data collected by a uniformly random policy over the entire

environment. With sufficient data the latent dynamics is accurate enough on most parts of the state space for control, therefore we do not observe a significant difference between online CARL and V-CARL. To further illustrate the advantage of V-CARL over online CARL, we modify the experimental setting by gathering initial samples only from a specific region of the environment (see Appendix E.1 for more details). Fig. 3 shows the learning curves of online CARL and V-CARL in this case. As expected, with biased data, both algorithms experience a certain level of performance degradation, yet, V-CARL clearly outperforms online CARL — this verifies our conjecture that control-aware LCE models are more robust to initial data distribution and superior in policy optimization.

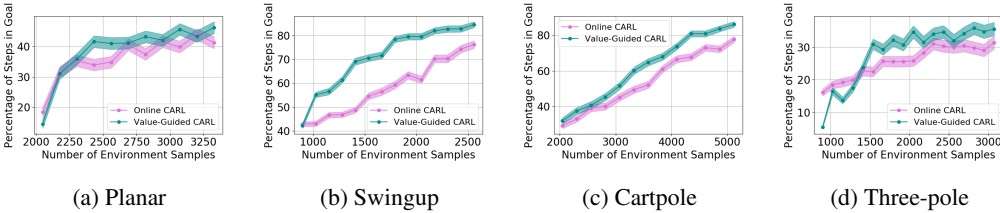

| (a) Planar | (b) Swingup | (c) Cartpole | (d) Three-pole |

Figure 3: Training curves of Online CARL and V-CARL with environment-biased initial samples.

## 6 CONCLUSIONS

In this paper, we argued for incorporating control in the representation learning process and for the interaction between control and representation learning in learning controllable embedding (LCE) algorithms. We proposed a LCE model called *control-aware representation learning* (CARL) that learns representations suitable for policy iteration (PI) style control algorithms. We proposed three implementations of CARL that combine representation learning with model-based soft actor-critic (SAC), as the controller, in offline and online fashions. In the third implementation, called *value-guided* CARL, we further included the control process in representation learning by optimizing a weighted version of the CARL loss function, in which the weights depend on the TD-error of the current policy. We evaluated the proposed algorithms on benchmark tasks and compared them with several LCE baselines. The experiments show the importance of SAC as the controller and of the online implementation. Future directions include **1)** investigating other PI-style algorithms in place of SAC, **2)** developing LCE models suitable for value iteration style algorithms, and **3)** identifying other forms of bias for learning an effective embedding and latent dynamics.

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
