# OpenReview forum: "Control-Aware Representations for Model-based Reinforcement Learning"
_ICLR.cc/2021/Conference — ICLR 2021 Poster_

### Official Review · AnonReviewer1 · 2020-10-17
**Good theoretical intuition but lacking experimentation**

**Rating:** 6
**Confidence:** 4

**Review:**

This paper examines the problem of learning controllable embedding (LCE), with the goal of learning good representations (usually achieved using variational inference algorithms) such that the maximum cumulative reward can be achieved. The main difference lies in the simultaneous learning of both the low-dimensional latent space as well as the action policy.

One of the main strengths of this paper is found in Theorem 1. The authors devise a simple policy iteration approach in the low dimensional learned space. Then, using mostly qualitative analysis, a bound on the policy improvement error is formulated. This error combines several intuitive and straightforward factors, which are then extracted to form more involved offline and online reinforcement learning algorithms. I have read through the proofs, and they seems correct.

While I appreciate the quality of the theoretical work, the paper had some drawbacks that brought me to my current score :
1. The loss function consists of many hyper-parameters. The authors should provide some guidelines for choosing these hyper-parameters due to the large number of possible combinations, and clearly state how the scalings affect performance.
2. Experimentation is lacking. While the authors conducted experiments mostly on toy problems, I expect them to compare against more involved environments which are harder to model. Their comparison with state of the art algorithms (e.g. Dreamer) which were also tested on such environments is thus not fair.
3. Minor comment: There is a newer version of Dreamer that the authors can compared against: https://arxiv.org/pdf/2010.02193.pdf
4. Minor comment: How would CARL compare against model-free offline RL methods, or generally to algorithms that are not SAC?

Question to authors:
Would there be a benefit in removing F altogether and learning a mapping X -> Z -> X’ without transitioning in the latent space? (i.e., errors III and IV in Theorem 1)

Finally, it would be beneficial if the authors could include code for their work. If the authors can't supply the complete code base, even code snippets with clarifying explanations to demonstrate their main ideas would be beneficial. This would greatly improve the quality and credibility of their work as well as the reviews.

To conclude, the paper provides strong theoretical intuition, which is a significant value-add of the paper. Nevertheless, its lack of experimentation and large number of hyper-parameters limit its overall quality. If the authors provide substantial improvement in the experimentation I will increase my score.

---

> ### Author Response · Authors · 2020-11-14
> **Response to R1**
>
> Response to main questions
>
> 1)  Theorem 1 provides a high-level guideline for selecting the hyper-parameters of the loss function: \lambda_{ed} = 2R_\max / (1-\gamma)^2, \lambda_c = \lambda_p = \sqrt{2} \gamma R_\max / (1-\gamma)^2, and \lambda_{reg} = \sqrt{2} \gamma R_\max / (1-\gamma). However, in practice, to further optimize the performance of the CARL algorithms, we set (a subset of) these hyper-parameters via grid search. To address the reviewer’s comment, we clarified this point by adding a discussion in Footnote 5.
>
> 2) It is important to note that we study the class of control problems in which the observations x have been selected such that the system is Markovian in the observation space X (see the beginning of Section 2). This is the same class of problems studied in E2C, RCE, SOLAR, and PCC, and this is why we selected them as our baselines. We did not include the results of E2C and RCE, because PCC has previously shown to be superior to them (see Levine et al. 2019). We compared our algorithms with Dreamer in X-Markovian problems considered in the paper, and Dreamer did not perform as well as CARL (with the same number of samples). This was expected as Dreamer has been designed for more general class of control problems (than X-Markovian), those that can be modeled as a POMDP. Extension of CARL to properly handle more involved environments (e.g., POMDP problems) requires using more powerful encoders (e.g., RNNs) and learning the latent reward function as a part of the representation learning process. As shown in Section 3, learning the reward function is in fact a part of the CARL’s loss function and can be easily included in the algorithm. Using other encoders requires a bit more work but we believe it should be doable. We left this extension and experimenting with more involved environments as a part of our future work.
>
> 3) We were not aware of this very new version of Dreamer and thank the reviewer for providing a reference to it. We added a citation to this work in the updated version of the paper.
>
> 4) As described in the paper, CARL is a model-based RL (MBRL) algorithm that works in the latent space. Similar to most comparisons between MBRL and model-free RL algorithms (e.g., see the MBPO paper), when the dynamics is learned reasonably accurately, CARL can be much more data-efficient than any model-free algorithm. However, we can definitely find problems in which a model-free algorithm outperforms CARL.
>
>
> “removing F”
>
> Removing F and learning the mapping X -> Z -> X’ is definitely a viable approach that definitely has the potential to be investigated as a future work. However, as described both in the paper and in Comment 4 about a model-free approach, these are all reasonable approaches that can work (or do not work) for some problems. However, the main idea of LCE (and as a result CARL) is to avoid direct prediction of the next observation, which could be challenging when the observation is high dimensional. Instead LCE suggests to learn a latent space and a latent dynamics F, and to control the systems there.
>
> “Code”
>
> The code is in PyTorch and we plan to open-source it with the final version of the paper.

---

### Official Review · AnonReviewer2 · 2020-10-28
**Official Blind Review #2**

**Rating:** 6
**Confidence:** 4

**Review:**

This paper proposes a new representation learning + RL algorithm called CARL, with a specific objective for learning a latent representation and dynamics model coupled with SAC policy learning in the latent space. Experiments on a few domains show CARL outperforming previous algorithms such as DREAMER and PCC.

Pros:

+ The key points of the paper are relatively well organized and motivated properly.
+ The experimental results succinctly demonstrate the promise of the proposed approach.

Cons:

- It is difficult to follow important details about the operation of this relatively complicated method.
- The experimental results are not sufficient for this largely empirical work.

With these pros and cons in mind, I am recommending a weak reject. See below for additional detailed comments.

EDIT: After discussion, I have increased my score and am recommending weak accept. See the discussion with the authors for details.


Quality
---

The paper studies an important problem, proposes a novel solution, and has promising experimental results. However, the main drawback in terms of the quality of the work is that the results are not complete enough. For work that is empirically driven, I do not view the current results as sufficient for publication.

In particular, DREAMER appears to be competitive with CARL on a few domains. But DREAMER was also evaluated much more broadly across many tasks from the DeepMind control suite, indicating a level of robustness and performance that is, at best, hinted at in this work for CARL. A wider suite of experiments, for example using the same tasks as DREAMER, would go a long way in better shaping the reader's understanding of the proposed method.

Clarity
---

As mentioned, the main points of the paper are presented well. The problem is properly motivated, and a central theorem gives rise to the proposed representation learning method. I did not check the proof for this theorem, but it appears sensible.

However, the finer points in the paper, which are also very important, are difficult to follow. For example, what is "model-based SAC"? There does not appear to be a proper explanation or citation for this. Is the learned dynamics model F used in some way to learn the Q-function? Is this novel, or is it from prior work?

Considering the proposition that replacing other control algorithms in the latent space with model-based SAC is important for the overall performance improvement, a description of model-based SAC is important. Furthermore, an ablation study would be helpful in terms of understanding the relative importance of this change vs the proposed representation learning approach, which seems to be the novel part.

Some other minor concerns about the methodological sections: there are many hyperparameters and not much guidance as to how to pick these; more discussion of why there are different versions of CARL and what are their respective strengths and weaknesses would be useful, especially for VCARL; I personally found the last paragraph of the VCARL description almost impossible to follow.

Originality
---

To the best of my knowledge, the representation learning algorithm itself is novel. Perhaps a related work that is overlooked is https://arxiv.org/abs/1907.00953, which apparently has been accepted to NeurIPS 2020 but has been out for some time. At a high level, this work also incorporates representation learning into SAC, though the underlying details are different. Still, this approach seems actually more closely related than some of the current citations and comparisons, e.g., SOLAR and DREAMER. At least a citation seems to be in order, and preferably a comparison. Indeed, this prior work also carries out a more comprehensive evaluation on more tasks than the current work.

Significance
---

This work has the potential to be significant, as many researchers and practitioners are currently interested in how to make deep RL more efficient and performative, in particular in visual settings. However, without a more comprehensive evaluation, it is difficult to judge for sure.

---

> ### Author Response · Authors · 2020-11-14
> **Response to R2**
>
> “more experiments”
>
> It is important to note that in addition to proposing algorithms to tackle the important problem of control from high-dimensional observations, we consider deriving a representation learning loss function from the control and dynamic programming principles as another contribution of our work. Returning to the sufficiency of our experiment, as explained in response to Reviewer 1, we study control problems in which the observations x have been selected such that the system is Markovian in the observation space X (see the beginning of Section 2). This is the same class of problems studied in E2C, RCE, SOLAR, and PCC, and this is why we selected them as our baselines, used the problems in their experiments, and conducted a comprehensive evaluation of these methods. We also experimented with Dreamer but as discussed in the paper, we did not expect it to perform well in our problems, because it has been designed for more general class of control problems (than X-Markovian), those that can be modeled as a POMDP. The goal of this paper is not to derive an algorithm that outperforms Dreamer (or similar algorithms) in problems that belong to the DeepMind suit. Our goal is to derive a representation learning loss function that is suitable for an important class of controllers (approximate policy iteration) and devise algorithms that properly interleave this representation learning and control. As explained in response to Reviewer 1, extending CARL to handle more involved problems requires using more powerful encoders (e.g., RNNs) and learning the latent reward function as a part of the representation learning process, which we believe both are doable. We left this extension and experimenting with more involved environments as a part of our future work.
>
>
> “model-based SAC”
>
> Model-based SAC is simply SAC when the data is generated from the model, instead of from the agent’s interaction with the environment. We thought that the meaning is clear, but to further clarify, we added a footnote (Footnote 6) to the updated version of the paper. However, we do not believe that this is enough reason for the reviewer to question the clarity of the paper and to state that important details are difficult to follow.
>
>
> “ablation study”
>
> We have done several ablation studies in the paper. Comparing offline CARL with PCC, because of the close connections between their loss functions, shows the importance of SAC as the control algorithm in place of iLQR in PCC. Comparing online CARL with offline CARL and PCC shows the importance of interleaving representation learning and control. Comparing online CARL with SOLAR shows the advantage of using the CARL loss function. Comparing CARL with and without policy distillation shows the effect of this process in the performance of the algorithm. Although there are other combinations that can be investigated, we believe we have already done a fair amount of ablation studies in the paper. If the reviewer has a particular ablation study in mind, it would be good to clearly state it that we see if we can provide its results by the end of the rebuttal phase.
>
>
> “hyper-parameters”
>
> Please see our response to Reviewer 1. To summarize, the theory (Theorem 1) provides a high-level guideline for selecting the hyper-parameters of CARL’s loss function. However, in practice, to further optimize the performance of the CARL algorithms, we set (a subset of) these hyper-parameters via grid search. To address the reviewer’s comment, we clarified this point by adding a discussion in Footnote 5.
>
>
> “why there are different versions of CARL”
>
> Offline CARL is for problems in which a large batch of exploratory data is available in advance, and thus, interleaving representation learning and control cannot add much value to the method. Moreover, it is used for an ablation study to compare CARL with PCC and see the effect of using SAC instead of iLQR. The main goal of V-CARL, as explained in the paper, is to establish a closer connection between representation learning and control by weighing the loss function using the TD-error of the current policy.
>
>
> “a related work”
>
> We thank the reviewer for bringing the SLAC work into our attention. It definitely has some connections to our work. We added a reference to it in the updated version of the paper.

---

> > ### Comment · AnonReviewer2 · 2020-11-17
> > **Main concerns still not resolved**
> >
> > Thanks for your response, it is helpful and does clear up some of my original questions and inquiries. It does not, however, adequately address my primary concern about the lack of a comprehensive empirical evaluation.
> >
> > As the authors point out, the empirical evaluation does not have to be viewed as the primary contribution. However, if the authors wish to present the theoretical result (and the loss function derived from it) as the primary contribution of the paper, the current state of the paper is still insufficient. The authors claim that comparing offline CARL with PCC is an ablation study. By my understanding of this work, I would disagree. Ablations are meant to isolate factors of variation in order to gain a deeper understanding of what changes and innovations contribute to observed differences. At least two things are varied between offline CARL and PCC: the loss function and the control algorithm. So, how important is newly proposed loss function vs the usage of model based SAC? I don't think that can be answered given the current experiments.
> >
> > This question, however, is crucially important. As I view it, if the authors wish to highlight the theoretical result as an important contribution, then the authors should provide concrete evidence that the resulting loss function actually contributes significantly to better performance. Otherwise, it is just a weak relationship between a theoretical lower bound and a loss function with a lot of hyperparameters. And to provide this evidence, the authors must actually ablate prior methods such as PCC by swapping in the new loss function, while keeping as many other components constant as possible. I.e., don't just throw in model based SAC as another improvement.
> >
> > However, the authors seem to think that the comparison between offline CARL and PCC actually "shows the importance of SAC as the control algorithm in place of iLQR", which would place the contribution as more empirical, which then comes back to my original point about needing more experiments. If this is the case, then again, in my own view, the theoretical result and proposed loss function do not have supporting evidence that they are actually important.
> >
> > Viewing this paper's contributions as a hybrid is also unsatisfying -- as I have discussed, currently neither the theoretical nor the empirical bits of the paper have enough evidence of their significance (though they are both promising).
> >
> > To round out my point about additional experiments: while I do not pretend that this would not require substantial time and compute, I do not agree that this is out of scope for this paper. If the observations need to be Markovian, just pass in multiple images from the past few time steps. This pretty much suffices for any of the tasks that DREAMER and SLAC experimented with. Reward prediction can be done by the dynamics model, as MBPO does it.
> >
> >
> > Some additional, more minor comments:
> >
> > In a similar vein to what is discussed above, I do not view comparisons between online CARL and PCC/SOLAR as ablations.
> >
> > Thanks for clarifying what is meant by "model based SAC". I'm not sure why the authors choose to cite the original SAC paper, which makes no mention whatsoever of models, rather than just directly citing the MBPO paper, since it sounds like the authors are basically doing exactly what MBPO prescribes? In my view, citing the original SAC paper does add confusion -- if a reader follows this citation, they will find a paper that never once mentions learning dynamics models, thus leading to ambiguity as to what the authors are actually doing.
> >
> > As a final note, I believe it is disingenuous, in the author response, to imply that my quip about model based SAC was "enough reason for [me] to question the clarity of the paper and to state that important details are difficult to follow". This was one example I brought up of several, including other points such as "there are many hyperparameters and not much guidance as to how to pick these" (thanks for adding a bit more detail about this) and "I personally found the last paragraph of the VCARL description almost impossible to follow" (left unaddressed in the author response).
> >
> > My score remains unchanged.

---

> > > ### Author Response · Authors · 2020-11-20
> > > **Thanks for the additional response**
> > >
> > > We thank the reviewer for reading our response and providing more comments. Here we try to further clarify some of the questions raised by the reviewer.
> > >
> > > “paper’s main contributions”
> > >
> > > We view the main contribution of the paper as deriving a loss function for representation learning from the principles of dynamic programming that is suitable to be used along with any (approximate) policy iteration algorithm. This is also reflected in the title of the paper, “control-aware representations ...”. In the supplementary materials (Appendix) of the original submission, we support this with offline and online implementations of the algorithm; ablation studies that show the effect of each term in the loss function (Appendix F2), the effect of policy distillation (Appendix F1), the effect of the control algorithm (Appendix F3); and a series of experiments that compare our algorithms with those that have been derived for the setting considered in the paper (E2C, PCC, SOLAR) on the domains used in these papers. The main algorithm of our paper is online CARL that extends E2C and PCC to an interactive (RL) setting, and our main theoretical result studies the effect of data generation distribution in LCE-style representation learning and control, which is novel and interesting as pointed out by Reviewers 1 and 3. Offline CARL is only a special case to better compare our framework with the previous algorithms like PCC. We also derived the V-CARL to see how much more we can bring the control algorithm (particularly the value function) to the representation learning process. In comparisons, PCC's loss for representation learning is derived solely based on studying an offline stochastic control problem, without considering any effects of data generation.
> > >
> > > “CARL Loss Function”
> > >
> > > The CARL’s loss function in the offline setting (offline CARL) has close connections to that of PCC, although they have been derived from different perspectives (one derived to be used with policy iteration style algorithms and one for locally linear control algorithms). The similarity between the loss functions of offline CARL and PCC is because they both have the prediction and consistency terms that according to our ablation studies (Appendix F2) are the most influential terms in CARL’s loss function. However, it is important to note that the loss function of online CARL has a significant difference with those of offline CARL and PCC, because it depends on the current policy (see Theorem 1) and not on state-action pairs. As discussed in Section 3 and also in our response to Reviewer 3, PCC and offline CARL have been designed for an offline setting (i.e., one-shot representation learning and control), and thus, the terms in their loss function are independent of a particular policy and are defined for state-action pairs (see the description of offline CARL in Section 4). On the other hand, online CARL has been designed for an online setting (interleaving representation learning and control), and thus, all the terms in its loss function depend on the current policy (see Theorem 1 and the description of online CARL in Section 4). This is a significant difference between the loss function of CARL and those of the previous algorithms that should not be ignored.
> > >
> > >
> > > “Ablation Studies”
> > >
> > > In the original main paper (Page 7, in the last line of the paragraph starting with "General Results"), we already pointed to the readers that our ablation studies are in Appendix F. We do not exactly know what kind of ablation studies the reviewer would like to see, in any case we should have referred that to the reviewer again but forgot to do so in our original response. In Appendix F1, we studied the effect of policy distillation (Line 4 of the Algorithm). We explained this in details in response to a question by Reviewer 3 about Line 4 of the algorithm. In Appendix F2, we studied the effect of each term in the CARL’s loss function and the results show the importance of the prediction and consistency terms, which is not that surprising. Finally in Appendix F3, we compare offline CARL and PCC. Because of the similarities between their loss functions, and given the fact that the prediction and consistency terms that they have in common happen to be the most influential terms in the loss function of offline CARL (ablation study of Appendix F2), we believe this comparison shows the effect of the control algorithms. The result suggests that SAC (an approximate policy iteration algorithm) performs better than iLQR (a locally linear controller) in the problems studied in the paper. Although other ablation studies can be done, ours focused on the loss function, control algorithm, and the effect of distillation, which we believe is a reasonable amount of ablation studies for a conference paper.

---

> > > ### Author Response · Authors · 2020-11-20
> > > **Thanks for the additional response (Cont'd)**
> > >
> > > “more experiments”
> > >
> > > We would like to emphasize that this is not a pure empirical paper (see our discussion about the main contributions). Moreover, we would like to reiterate our argument that extending CARL to more involved problems (e.g., those in the DM suite) requires using more powerful encoders. We believe this extension is doable but requires a separate, dedicated work to extend our theoretical results and loss function to that case, which is an interesting future work. The reviewer refers to (i) Dreamer and SLAC and (ii) stacking frames to turn the problem Markovian. Dreamer and SLAC use sequence-to-sequence encoding, which is different than our formulation that maps observations to a single latent state. Our theoretical results currently do not support this type of advanced encoding mechanism. We already stack frames, but only a few of them, to turn our problems Markovian. When the number of frames required to turn the problem Markovian is large, we need more powerful encoders, such as the recurrent NN based encoder used by the above methods.
> > >
> > > “Paper’s Clarity”
> > >
> > > We hope Footnotes 5 and 6 that we added to the updated version of the paper have addressed the reviewer’s concern regarding model-based SAC and the hyper-parameters of the algorithm. We will also revise the last paragraph of the VCARL description to make it more accessible. If there is something specific about this paragraph that the reviewer found difficult, please let us know to address it specifically, given the space limits of the paper.

---

> > > > ### Comment · AnonReviewer2 · 2020-11-23
> > > > **Thanks, this slightly improves my overall assessment**
> > > >
> > > > Thanks for providing additional comments, I believe this clarifies some of my confusion. I now better understand the significance of the theoretical bits, which result in a similar loss function as PCC but is motivated and derived from a different and more general perspective. I still believe that the link between the theorem and the actual loss function is weak, as evidenced by the number of hyperparameters that have to be searched over. But this is acceptable to me.
> > > >
> > > > The way I (now) view it, the resulting loss function is primarily interesting and significant because it allows for seemingly easier integration of general model based policy iteration algorithms. But we are only interested in this because policy iteration algorithms seem like they should work better in practice on a larger variety of tasks. Therefore, the primary contribution of this paper is still empirical, and the empirical evaluation is still lacking.
> > > >
> > > > I still do not follow the authors' justification about not being able to run additional experiments. The authors clarify that they are indeed stacking images to produce Markovian observations. If this is the case, it should be even simpler to try more tasks such as DM control suite. Why would the number of images that need to be stacked be larger for other tasks? If the actions are torque control, then velocity is the missing information that is not present in one image, but can be inferred from a stack of two images. In practice, stacking just a few images should be enough for every task visualized in the [DREAMER paper](https://arxiv.org/pdf/1912.01603.pdf) Fig 2.
> > > >
> > > > Anyway, I'm happy to discuss further with the authors if they wish, but that is up to them, as I am increasing my overall score from weak reject to weak accept.

---

> > > > > ### Author Response · Authors · 2020-11-25
> > > > > **Thanks for your feedback, more response to your questions**
> > > > >
> > > > > We would like to thank the reviewer for quick reaction to our responses, asking questions, providing feedback, and being open to conversation. We hope we have been able to address the issues raised by the reviewer and to clarify their questions. Here are a few points in response to the reviewer’s latest post. Hope they further clarify different aspects of our work.
> > > > >
> > > > >
> > > > > “similarties between CARL and PCC’s loss functions”
> > > > >
> > > > > We would like to emphasize there is similarity between the loss functions of PCC and offline CARL. The loss function of online CARL is significantly different because it depends on the current policy and not on state-action pairs.
> > > > >
> > > > >
> > > > > “number of hyper-parameters”
> > > > >
> > > > > CARL has 5 hyper-parameters, but we only fine-tune 2 of them (prediction and consistency), because our ablation studies in Appendix F suggest that these two play the most important roles in the performance of the algorithm. All other algorithms in this area (E2C, PCC, Dreamer) have the same number of hyper-parameters (the weights of the different terms in their loss function). They all have to tune 2-3 hyper-parameters. So, in that sense, we do not see much difference between CARL and its counterparts.
> > > > >
> > > > >
> > > > > “primary contribution”
> > > > >
> > > > > We see our primary contribution as both algorithmic and empirical. Our algorithmic contribution includes deriving a loss function from the principles of dynamic programming for learning representations that are suitable for a large class of control algorithms, namely approximate policy iteration algorithms, and three algorithms that use different forms of this loss function with SAC in offline and online settings. Our empirical contribution includes a number of ablation studies, comparing our algorithms with those (PCC, E2C, RCE) that have been derived for the setting considered in the paper and on the same domains used in these papers, and comparing them with Dreamer.
> > > > >
> > > > >
> > > > > “experiment with DM control suit”
> > > > >
> > > > > The number of images needed to be stacked for the tasks in the DM control suite is larger than those in our experiments because of the colored image, where each input observation for these tasks should be multiplied by 3. This is why we believe we need to combine CARL with more powerful encoders to handle these tasks. In any case, we used the black and white version of 4 tasks (Pendulum, Acrobot, Cart-Pole, and Cart-k-Pole) in the DM control suite in our experiments. We will try our best to convert Hopper and Walker to black-and-white and include the results of applying CARL to these tasks in the final version of the paper.

---

### Official Review · AnonReviewer3 · 2020-10-30
**The algorithm is promising but the theoretical foundation is inaccurate**

**Rating:** 6
**Confidence:** 4

**Review:**

This paper aims to address an important question in reinforcement learning: policy learning from high-dimensional sensory observations. The authors propose an algorithm for Learning Controllable Embedding (LCE) based on policy iteration in the latent space. The authors provide a theorem to show how the policy performance in latent-space policy improvement depends on the learned representation and develop three algorithmic variations that attempt to maximize the theoretical lower bounds. In the experiments, the proposed algorithm CARL shows improved performance when compared with other LCE baseline algorithms.

While I'm not particularly familiar with the field of LCE, I think the idea of learning a representation that is suitable for policy improvement is an interesting idea. The readability of this paper is also pretty good, which can be difficult to get right because the of the correspondence between the original space and the latent space. Overall the paper is easy to follow.

While I do think Algorithm 1 is reasonable, I found its theoretical foundation, namely Theorem 1, is incorrect. In the proof of Theorem 7 on p15 in the appendix, I do not think the implication T^2 VE(x) < T VE(x) + \gamma Delta(x) for all x, would hold. Because Bellman operator contracts in the L-inf norm, a basic inequality would rather take a form of  T^2 VE(x) < T VE(x) + \gamma sup_y Delta(y). In addition to this, another minor error happens in the first equation on pg 16, where I believe the correct right hand side would be 1/(1-gamma) sup_y Delta(y), without the gamma dependency.

However, a bound that depends on L-inf norm would be quite bad for Theorem 1, and current data collection process in Alg 1 is not sufficient for minimizing it. I think it might be possible not using an L-inf bound but using an expected error based on the policy's rollout distribution. However, this change would largely change the theoretical results, and perhaps the motivation or details of the algorithm design. Therefore, I do not think the paper is ready for acceptance at the current stage without a large revision. If the authors can address this question properly, I would raise my score.

Beyond the flaw in the theory, there are some parts which can benefit from some clarification:
1. In the offline CARL, how does the algorithm address the issue of out of distribution error due to using a batch dataset?
2. The authors argue that the loss here is different from PCC many times in the paper, but they never explain whether the choice here is better (or in which way).
3. In line 4 of Alg 1, how do we ensure such pi would exist?
4. What is the definition of "compatible reward function" in the last paragraph on p4?
5. For completeness of presentation, please include the definition of curvature loss.

---

> ### Author Response · Authors · 2020-11-14
> **Response to R3**
>
> “Proof of Theorem 7”
>
> We thank the reviewer for bringing into our attention this major typo in the proof of Theorem 7, and as a result in the statement of Theorem 1 (Eq. 4). You are right, when we apply the Bellman operator to TV - V < \Delta, we do not obtain T^2V(x) - TV(x) < \gamma \Delta(x), for any x. However, we can show that for any x, we have T^2V(x) - TV(x) < \gamma E_{x\sim P_{\pi o E}}][\Delta(x)]). This will result in a change in the statement of Theorem 7 and all the subsequent results, and finally in the statement of Theorem 1, that x should come from the \gamma-occupancy measure of the current policy (\pi o E). So, we do not obtain a result in L-inf norm that as correctly mentioned by the reviewer is not desirable. We revised the appendix and the statement of Theorem 1 (Eq. 4) in the paper to reflect this change. However, this change has no effect on the CARL algorithms as the samples are collected by following the current policy (\pi o E).
>
> Response to other questions
> 1) In Offline CARL, similar to other LCE methods, such as E2C and PCC, we assume that the data is collected by an exploratory policy that provides a good coverage of the parts of the state-action space that are relevant to the task at hand (see the discussions on Page 5). Violation of this assumption would add error to the process, similar to all offline RL settings, and require certain corrections to alleviate its effects. Studying this issue is outside the scope of this work and is an interesting future direction.
>
> 2) As mentioned in the paper, the CARL’s loss function has close connections to that in PCC (see Page 5), although they have been derived from completely different perspectives. While the loss function in CARL has been derived such that the learned representation is suitable for a policy iteration style algorithm, the one in PCC has been derived to learn a latent space that is amenable to locally linear control algorithms. As discussed in Section 3, since PCC has been designed for an offline setting (i.e., one-shot representation learning and control), its prediction and consistency terms are independent of a particular policy and are defined for state-action pairs. While CARL has been designed for an online setting (i.e., interleaving representation learning and control), and thus, all its loss terms depend on the current policy. Despite the similarities in loss functions, as we show in our experiments, offline CARL (which is a member of the CARL family) outperforms PCC. Moreover, the other members of the CARL family are more superior to PCC and offline CARL, as shown in our experiments, mainly because they better address the data collection issue discussed in Question 1, by interleaving representation learning and control.
>
> 3) As discussed in the description of online CARL in Section 4, we approximate the operation in Line 4 of Algorithm 1 by a process we refer to as policy distillation. To compute \pi, we project the observation policy \mu onto the latent space by minimizing D_KL(\mu || \pi \circ E). There might be other ways to implement this line of the algorithm (including removing it), which requires more investigation. We showed in our experiments (see Appendix F1) that the results when we remove this line (no distillation) are worse than those with distillation. We revised the paper to better explain this step of the algorithm.
>
> 4) Similar to other LCE methods, such as E2C and PCC, we define the reward function in the latent space. We refer to a reward function as compatible, if when it is optimized (in the latent space), the resulting policy (projected back to the observation space) solves the problem. For example, in a goal-based task, a reward function that measures the negative distance from each latent state to the image of the goal (in the latent space) is compatible. For clarification, we added the above description to the paper (see Footnote 4).
>
> 5) We added the definition of the curvature loss, which is identical to that in the PCC paper, at the end of Section 3 (see Page 5).

---

### Author Response · Authors · 2020-11-14
**Rebuttal**

We thank the reviewers for their useful feedback. Please see response and the updated paper to the individual comments below.

---

### Decision · Program_Chairs · 2021-01-07
**Final Decision**

**Decision:**

Accept (Poster)

**Comment:**

This paper addresses the question of RL in high-dimensional spaces by learning lower-dimensional representations for control purposes. The work contains both theoretical and empirical results that shows the promise of the proposed approach.

While the reviewers had initial concerns, including with a problem in a proof and questions around the contributions, after robust responses and discussions this paper is now in good shape.